

# Attendance, engagement and performance in a medical school curriculum: early findings from competency-based progress testing in a new medical school curriculum

Heather S. Laird-Fick[1], David J. Solomon[2], Carol J. Parker[2,3] and Ling Wang[1]

[1] Department of Internal Medicine, Michigan State University, East Lansing, MI, United States of America
[2] Office of Medical Education Research and Development, Michigan State University, East Lansing, MI, United States of America
[3] Academic Affairs, Michigan State University, East Lansing, MI, United States of America

Corresponding author
David J. Solomon,
dsolomon@msu.edu

## ABSTRACT

**Introduction**. Medical students often do not value attending in-person large group sessions. It is also not clear from prior research whether attendance at large group sessions impact on performance in medical school. The goal of this study was to assess the relationship between voluntary attendance in large group sessions organized as a "flipped classroom" in a new innovative curriculum and students' mastery of clinical applications of basic science knowledge.

**Methodology**. Our students' ability to apply basic science knowledge to clinical problems is assessed via progress testing using three methodologies: a locally developed multiple-choice examination, written examination developed through the National Board of Medical Examiners (NBME) Customized Assessment Services Program and post encounter questions included in a clinical skills examination. We analyzed the relationship between voluntary attendance at weekly large group "flipped classroom" sessions and the students' performance on examinations given at four intervals over the initial 24-week module of the medical school curriculum.

**Results**. Complete data were available for 167 students. A total of 82 students (49.1%) attended all large group sessions, 65 students (38.9%) missed one or two sessions and 20 students (12.0%) missed three or more sessions. There were no difference between the students in the groups on their medical admission (MCAT) examination scores. The growth in performance from each time point until the next was statistically significant. There was no statistically significant difference in growth between the students who had no absences and those who had one or two absences. Students who missed three or more sessions performed significantly lower than their peers over the 24 week module and were more likely to score one or more standard deviations below the class mean on the assessments.

**Conclusions**. We found no relationship between attendance and MCAT scores suggesting the differences in performance on the progress tests was not due to initial differences in knowledge or reasoning skills. While the study was not experimental, it suggests large group sessions using a "flipped classroom" approach to provide reinforcement, feedback and practice may be effective for increasing learning and retention in the application of basic science knowledge among first year medical students.

## BACKGROUND

In the fall of 2016 Michigan State University's (MSU) College of Human Medicine (CHM) implemented a new curriculum called the Shared Discovery Curriculum (SDC). The SDC is organized around patient chief complaints and concerns and student assessment is done via progress testing. Progress testing uses parallel forms of a very broad-based examination given multiple times over an extended course of study and is well suited for problem-based learning curricula such as the SDC (*Van Der Vlueten, Verwijnen & Wijnen, 1996*). A detailed description of the SDC curriculum and the Progress Suite of Assessments are available on the SDC website (*MSU, 2015*).

The Early Clinical Experience (ECE) constitutes the initial 24-week module of the SDC. Students are trained in basic data gathering and patient communication skills and after eight weeks begin working in clinic settings with medical assistants and nurses. They also begin mastering clinical applications of basic science knowledge through guided independent study, problem-solving exercises in small groups, and in weekly large group sessions. The large group sessions use a flipped classroom model, during which the students apply basic science concepts they studied over the preceding three days. Attendance at the large group session was encouraged but not required during the first year the curriculum was implemented.

Medical students do not always see the value of attending large group sessions given the increasing availability of educational resources designed to help them prepare for the United States Medical Licensure Examination (USMLE) Step 1 (*Zazulia & Goldhoff, 2014*). Research on the relationship between classroom attendance and academic performance in medical school is mixed. *Fogleman & Cleghorn (1983)* reported an association between self-reported class attendance and National Board of Medical Examiner (NBME) Part I performance. Other studies in health education, however, have not found a relationship between attendance and performance (*Azab et al., 2015*; *Eisen et al., 2015*). The relationship between attendance and grades among undergraduate college students however was found to be quite strong and relatively independent of other predictors in a large scale meta-analysis. (*Credé, Roch & Kieszczynka, 2010*) The lack of consistent findings in health professions education could be do to a number of factors. There are relatively few studies, most small scale and with differing conditions.

As part of the evaluation of the SDC we assessed the relationship between attendance in the large group sessions of the ECE and performance on the progress tests given over the first year of the curriculum. The goal of this study is to assess the relationship between voluntary attendance in the ECE large group sessions and the students' mastery of clinical applications of basic science knowledge in the SDC and help determine whether it would be prudent to require attendance in these sessions in the future.

## METHODS

The large-group activities (LGAs) employed a flipped classroom approach. Preparatory materials were provided to the students three days prior to an LGA session. To assess how well the students had prepared for the LGA, they took an Individual Readiness Assessment Test (IRAT) at the beginning of each LGA session. The IRAT was used to provide formative feedback and not for grading purposes. We used a score of zero on the IRAT as a proxy measure to identify students that did not attend the large group session that week. Each IRAT had between 28 and 50 points and it is highly unlikely a student who completed the test would have a legitimate score of zero. Students were categorized into three groups: (1) those who attended all sessions, (2) those who missed one or two sessions and (3) those who missed three or more sessions. With 24 sessions over the course of the ECE, we hypothesize it would be quite possible for a student who intended to attend all the large group sessions to miss one or two sessions for reasons largely beyond their control, such as an illness or family emergency. We also hypothesized that when students missed more than two sessions they probably do so as a conscious decision that their time would be better spent in other activities rather than for reasons beyond their control. Given that each LGA focused on the application, rather than the delivery, of new content, the students' primary penalty for missing an LGA would be to miss an opportunity to receive additional practice and formative feedback on the concepts covered that week.

As part of the Progress Suite of Assessments (PSA), students' ability to apply basic science knowledge to clinical problems is assessed using three methodologies. Students complete the PSA at four points during the 24-week ECE; shortly after matriculation in late September, mid-November, mid-February of the following year and at the end of the ECE in April of the following year.

The first assessment methodology is a multiple-choice examination that is locally developed and consists of 140 items from a large item pool used in the previous curriculum as well as additional items created to ensure comprehensiveness. While each of the four administrations included different items, they were designed to be roughly equivalent though they were not equated through a formal psychometric process. The second PSA was taken at about the same interval and was designed through the National Board of Medical Examiners (NBME) Customized Assessment Services Program (http://www.nbme.org/schools/cas.html). While slightly different domains of content were included in each of the four administrations, we have treated the different versions as roughly equivalent examinations. Along with written examinations, students take a Progress Clinical Skills Examination (PCSE) that includes a post encounter component designed to assess students' ability to apply basic science knowledge to the standardized patient interaction they just completed. There were eight cases in each PCSE examination. The number and nature of the items varied from case to case. For example, the student may have to write a note about the patient or answer some questions about the basic science underlying the patient's complaint. Gold and his colleagues describe the PCSE in more detail (*Gold et al., 2015*). The percentage of items correct was used as the outcome

measure in both sets of written examinations. The percentage of possible points that could be achieved in the post encounter stations was used as the outcome measure for the post encounter stations.

We assessed the relationship between large group attendance and the three performance measures in two ways. First, we conducted a repeated measure analysis of variance (ANOVA) for the three PSAs with the four administrations of these assessments as the design over measures and the categorized number of missed large group sessions as the design over subjects. We used planned comparisons to test for differences in the levels of each factor. For the design over measures we tested the significance of the change in score from the first administration to the second, the second to the third and the third to the fourth. For the absence categories, we tested for a difference between students with no absences and students with one or two absences and then for a difference between students with two or fewer absences and students with three or more absences.

Secondly, we compared the number of times a student scored at least one standard deviation (SD) below the mean for each of the three measures across all four time points (total of 12 assessments) for the three groups of students. We used a score of one SD or more below the mean as an indicator that a student's performance was well below their peers on that assessment and may need remediation. We categorized this measure into students who did not have any scores at least one SD below the mean, those who had one to four scores one or more SD below the mean and those who had more than four scores one or more SD below the mean. The relationship between the categorized number of absences and categorized number of assessments where the student scored one or more SD below the mean was tested by Fisher's exact test. As with the repeated measures we conducted two comparisons, students without any absences with those who had one or two absences; and students who had less than three absences with those who had three or more absences.

We also compared the students' Medical College Admissions Test (MCAT) total scores among the students in the three absence groups. The exam is a written multiple-choice test taken by medical school applicants that assesses critical analysis and reasoning skills along with relevant basic science and behavioral knowledge. We used a one-way analysis of variance to test for statistically significant differences among the students in the three attendance groups.

We used SPSS Version 24 to conduct the statistical analyses and considered statistical tests where $p < 0.01$ as statistically significant. An "honest broker" was used to provide deidentified data to conduct this study. The use of an honest broker to provide deidentified and hence not human subject data has been approved by the Michigan State University Research Protection Program.

## RESULTS

Complete data were available for 167 students. A total of 82 students (49.1%) attended all large group sessions, 65 students (38.9%) missed one or two sessions and 20 students (12.0%) missed three or more sessions. We compared the MCAT scores of the three groups above to assess if they entered medical school with different levels of basic/behavioral knowledge or reasoning skills. We found essentially no differences in MCAT performance

**Table 1  Progress suite of assessment performance by number of missed large group sessions.**

| | | NBME | | Local exam | | Post encounter | | |
|---|---|---|---|---|---|---|---|---|
| | | Mean | Std. Deviation | Mean | Std. Deviation | Mean | Std. Deviation | N |
| No missed sessions | Sept 2016 | 37.17 | 4.43 | 38.56 | 4.90 | 35.73 | 7.14 | |
| | Nov 2016 | 41.80 | 5.36 | 39.46 | 4.72 | 40.05 | 5.17 | 82 |
| | Feb 2017 | 49.07 | 5.93 | 44.48 | 5.77 | 48.32 | 5.91 | |
| | Apr 2017 | 53.27 | 6.52 | 47.27 | 5.75 | 49.93 | 6.72 | |
| One or two missed sessions | Sept 2016 | 36.75 | 5.30 | 37.49 | 4.70 | 36.69 | 8.39 | |
| | Nov 2016 | 40.37 | 6.53 | 39.47 | 6.46 | 38.20 | 7.85 | 65 |
| | Feb 2017 | 48.23 | 7.98 | 44.98 | 7.06 | 46.46 | 5.51 | |
| | Apr 2017 | 52.86 | 9.34 | 46.77 | 7.91 | 49.34 | 7.19 | |
| Three or more missed sessions | Sept 2016 | 33.80 | 6.11 | 34.64 | 5.49 | 35.20 | 8.70 | |
| | Nov 2016 | 35.40 | 8.09 | 35.96 | 5.83 | 35.60 | 6.03 | 20 |
| | Feb 2017 | 42.20 | 8.14 | 39.46 | 5.38 | 45.05 | 6.30 | |
| | Apr 2017 | 48.20 | 9.20 | 41.78 | 7.89 | 45.00 | 8.54 | |

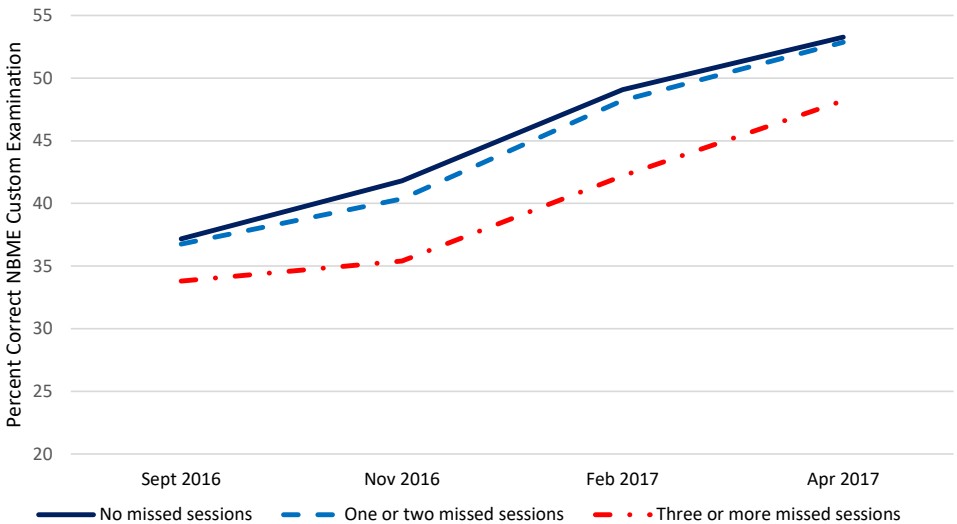

**Figure 1  NBME performance by number of large group absences.**

among the students in the three groups. The total MCAT scores were 506.48 ± 5.00, 506.44 ± 5.54, and 506.56 ± 5.16 (mean ± SD) for the group with no missed sessions, one or two missed sessions and three or more missed sessions. The differences were not statistically significant.

Table 1 presents summary statistics over the four administrations and within the three absence categories for the NBME customized examination, locally developed examination, and the post encounter component of the PCSE. Figures 1–3 present these data graphically. Based on the repeated measures ANOVAs, there was no statistically significant interaction between the categorized number of absences and the change over time in student performance for any of the PSAs although the interaction for the NBME and the post

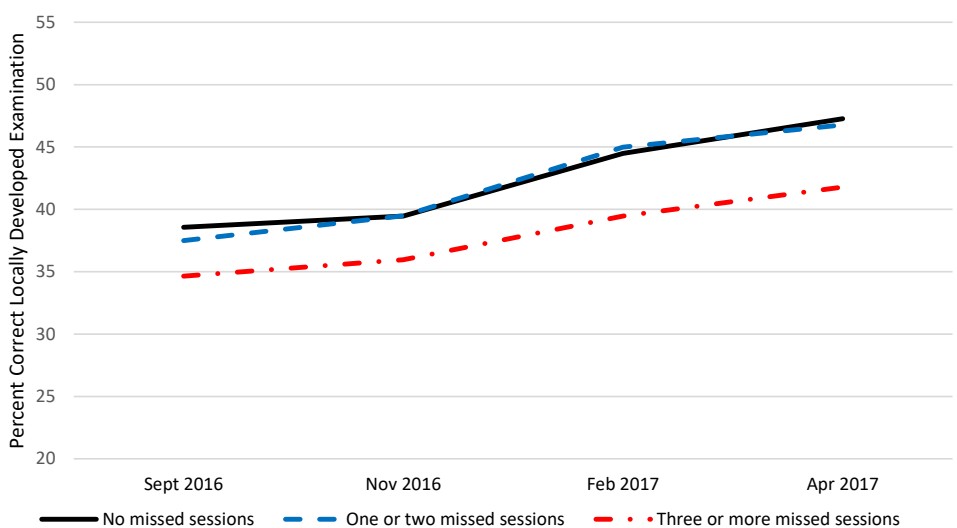

**Figure 2** Locally developed performance assessment by number of large group absences.

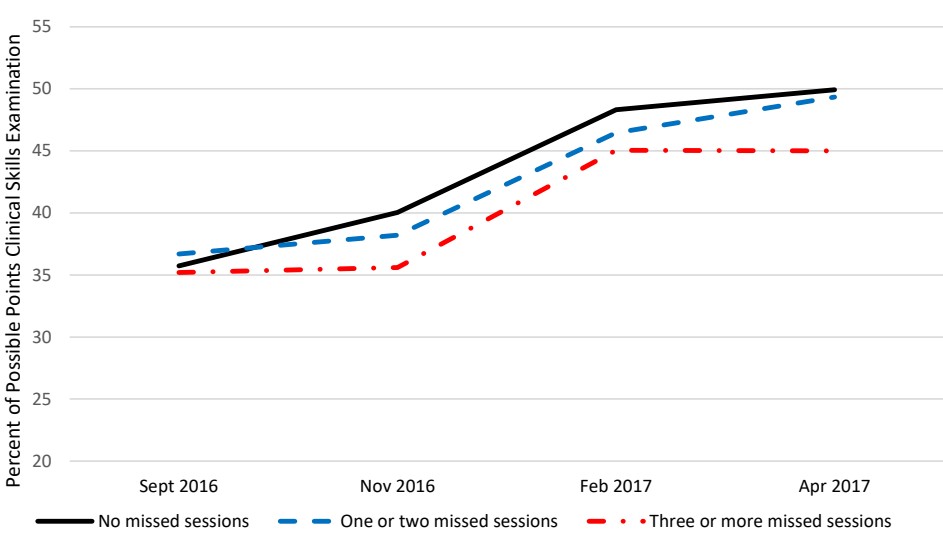

**Figure 3** Clinical skills exam post encounter station performance by number of large group absences.

encounter component of the PCSE approached statistical significance at ($p = 0.02$) and ($p = 0.03$) respectively. For all three PSAs, the growth in performance from each time point until the next was statistically significant ($p < 0.01$) except for the post encounter component of the PCSE where the change Feb. 2017 to Apr. 2017 approached statistical significance ($p = 0.03$). Also, for all three PSAs, there was no statistically significant difference between the students who had no absences and those who had one or two absences. There was a statistically significant difference ($p < 0.01$) between students who

**Table 2  Number of assessments at least one SD below the mean by number of missed large group sessions.**

| | Categorized number of scores 1 SD below the mean | | | |
|---|---|---|---|---|
| | No missed sessions | 1 or 2 missed sessions | 3 or more missed sessions | Total |
| None | 45 | 24 | 2 | 71 |
| | 54.9% | 36.9% | 10.0% | 42.5% |
| 1 through 4 | 30 | 29 | 13 | 72 |
| | 36.6% | 44.6% | 65.0% | 43.1% |
| More than 4 | 7 | 12 | 5 | 24 |
| | 8.50% | 18.50% | 25.00% | 14.40% |
| Total | 82 | 65 | 20 | 167 |
| | 100.0% | 100.0% | 100.0% | 100.0% |

**Notes.**
Statistical significance based on Fisher's exact test.*
No missed sessions vs one or two missed sessions $p = 0.54$.
Less than three missed sessions vs 3 or more missed sessions $p < 0.01$.

had two or fewer absences and those who had three or more absences with the students with fewer or no absences performing significantly better than those with three or more absences.

Table 2 presents a crosstabulation of the categorized number of times a student scored one SD or more below the mean by the categorized number of absences. We found no statistically significant difference in the categorized number of scores one or more SD below the mean for students with no absences and those with one or two absences. Students with three or more absences had more scores that were one or more SD below the mean than students with two or less absences ($p < 0.01$).

## DISCUSSION

Our students' performance on the three PSAs improved significantly over the 24-week ECE experience. Given this is the first year of a newly implemented curriculum these findings are encouraging. Students who chose to attend 22 or more of the 24 weekly large group sessions performed better on all three PSAs and were considerably less likely to receive scores one or more SD below the mean when compared with students who had three or more large group session absences. These differences persisted across the 24-week module. While students with higher attendance rates out performed students with lower attendance, there was no differences in MCAT scores among the students in the three attendance groups. This suggests the differences in performance on PSAs do not reflect initial differences in basic and behavioral science knowledge or reasoning skills which the MCAT is designed to measure. There also was no interaction between growth over time and absences suggesting that while students with three or more absences did not performed as well as students with less than three absences initially, the gap did not widen or narrow at least to the point where it was statistically significant.

The large group sessions used a "flipped classroom" approach where new material was not introduced. They instead provided reinforcement, feedback and practice for the material already introduced to the students in their individual and small group curriculum.

There is evidence that this approach to curriculum design my be more effective than the traditional teacher-centered approach to large group instruction for learning essential basic science material (*Street, Gilliland & McNeil C. Royal, 2015*) as well as increasing student satisfaction and engagement in the curriculum (*McLaughlin et al., 2014*). This approach is also consistent with a social constructivist perspective on learning which emphasizes the importance of discussion and interaction with other students and teachers in the learning process. (*Palinscar, 1998*).

There is not a large body of research on the relationship between class attendance and performance in medical school and the findings are mixed in the few studies done. More research has been done on students at the undergraduate college level. *Credé, Roch & Kieszczynka (2010)* conducted a meta-analysis of undergraduate college students that provides a comprehensive summary of a large body of research on the relationship between class attendance, grades and other student characteristics. It is not clear whether their finding can be applied to medical students, but it seems reasonable to consider their finding in interpreting the results of our study. As in our study, Credé and his colleagues (*2010*) found a moderate relationship between attendance and performance. In their meta-analysis attendance correlated $p = 0.44$ with course GPA and $p = 0.41$ with overall college GPA which was more predictive than SAT scores, high school grades or surveys of study habits/skills.

Since our study was not experimental we cannot be sure the relationship between attendance in the large group sessions and performance on the PSAs is causal. The relationship between attendance and the PSAs may be mediated by some third factor or factors such as conscientiousness, motivation and/or study skills/habits. Credé and his colleagues' (*2010*) meta-analysis suggested this was not the case for the relationship between class attendance and grades in undergraduate students. While it is not clear the relationship between attendance in the large sessions and performance on the PSAs is causal, the evidence was a major consideration in our medical school's decision to start requiring students to attend the large group sessions in the ECE. It should also be noted that this study was based on a single year's worth of data and this was the first year of a new curriculum.

## CONCLUSIONS

Our study found that students who voluntarily attended all or most large group "flipped classroom" teaching sessions in the first 24-week block of our medical school curriculum performed significantly better on progress skills assessment examinations. We found no relationship between attendance and MCAT scores suggesting the differences in performance on the PSAs was not due to initial differences in knowledge or reasoning skills which the MCAT is designed to assess. While the study was not experimental, it suggests large group sessions using a "flipped classroom" approach to provide reinforcement, feedback and practice may be effective for increasing learning and retention among first year medical students.

### Funding
The authors received no funding for this work.

### Competing Interests
The authors declare there are no competing interests.

### Author Contributions

- Heather S. Laird-Fick and Carol J. Parker conceived and designed the experiments, performed the experiments, authored or reviewed drafts of the paper, approved the final draft.
- David J. Solomon conceived and designed the experiments, performed the experiments, analyzed the data, prepared figures and/or tables, authored or reviewed drafts of the paper, approved the final draft.
- Ling Wang conceived and designed the experiments, performed the experiments, analyzed the data, authored or reviewed drafts of the paper, approved the final draft.

### Ethics
The following information was supplied relating to ethical approvals (i.e., approving body and any reference numbers):

The Michigan State University Research Protection Program approved this study.

### Data Availability
The raw data are provided in a Supplemental File.

### Supplemental Information
Supplemental information for this article can be found online at http://dx.doi.org/10.7717/peerj.5283#supplemental-information.

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
