# Peer review of "Attendance, engagement and performance in a medical school curriculum: early findings from competency-based progress testing in a new medical school curriculum"

_PeerJ, doi:10.7717/peerj.5283_

## Round 0.1 · original submission · Minor Revisions

Dear David,

I am looking forward to receiving a new version of your manuscript revised according to the reviewers' comments.

Best regards,

Yoshi

Reviewer 1 ·

Basic reporting

The manuscript is clear and professional in its use of English and structure. The background provides sufficient depth to evaluate how the manuscript fits into the broader field of knowledge. Although increasingly used in health professions schools, more research is needed into the best practices of flipped-classroom implementation. This paper contributes to this field in addressing the effects of student attendance with respect to performance.

Specific suggestions, in order of importance, the authors should consider in this section of the review include the following.

1. Lines 12-13: It would be helpful to provide more detail with respect to the other parts of the ECE. For example, how long are the large group sessions? How long are the problem-solving sessions? When do the problem-solving sessions happen relative to the weekly large group sessions? What is the relationship between the preparatory materials, and the activities in the problem-solving session and in the weekly large group sessions? How do the problem-solving sessions differ from the large group sessions in terms of the student activities? What is the attendance requirement for the problem-solving sessions? In the large group sessions, are students assigned to fixed groups or do they self-select?
2. Line 7 and elsewhere: According to instructions to author, “PeerJ uses the "Name. Year" style with an alphabetized reference list.” Thus, the superscripted numbers for the reference should be revised as per the instructions and the reference list alphabetized.
3. Line 7: The abbreviation PBL in parentheses is provided but the abbreviation is not used in the text subsequently. It can be deleted.

Experimental design

The experimental design, the assignment of students to cohorts, and the statistical analyses were all appropriate.

Specific comments, in order of importance:
1. Line 112: “Tables” should read “Table”
2. Lines 115-117: The indicated p value, p<0.01, should read p>0.01 since there was no statistically significant interaction.
3. Line 38: Did the IRAT count toward their grade in the ECE or was it just based on the progress test? How did the students who attended the sessions perform on the IRAT?
4. Lines 45-46: The authors hypothesize on why students missed class? Did students complete an evaluation of the ECE at the end? If so, were there any comments which can provide insight into why students did not attend class? It may be interesting to conduct focus groups with students who were absent.
5. Line 58: The authors state the locally developed exams were “roughly equivalent.” Some testing programs (e.g., ExamSoft) provide an overall difficulty for an exam based on past performance of the questions. It may be interesting to know whether the difficulty of each exam was comparable. This will not affect the overall results and conclusions but would offer a more declarative statement with respect to equivalency of the exams.
6. Lines 89-90: The authors state, “students who had two or less absences.” Just to be clear, does this mean that the data for the students without any absences were grouped with those who had one or two absences?

Validity of the findings

The raw data were provided and the results of the analyses were logically presented. The discussion aligned well with the results presented.
The discussion section could be enhanced, however, by some additional discussion of the effects of flipped-classroom on student performance with speculation on why absence from these sessions could lead to poorer student performance. For example, Street, et al. (Med Sci Educ 2015; 25:35) provides evidence in support of enhanced learning in the flipped classroom. Results from other health professions may be germane including Physical Therapy 2001; 896, Acad Med. 2014; 89:236, Am J Pharmaceut Educ. 2014; 78, and Adv Physiol Educ 32013; 7:316. Additionally, since the journal allows for speculation as long as it is identified as such, it may be useful to the reader to briefly mention the socio-constructivist view of learning which argues that knowledge is socially constructed through interaction, engagement and participation and that, pedagogically, the flipped classroom creates these social learning opportunities. It could be speculated that students who decide not to come to class are missing out on this social learning opportunity. The work of Vygotsky is relevant here.

Additional comments

Curriculum reform is increasingly occurring in medical schools. As a result, there are consequences, some intended, some not, of such reform. Thus, it is critical that schools evaluate the effects of the curriculum change on student performance to examine the consequences. This paper is timely in that the authors summarize briefly their new curriculum, the Shared Discovery Curriculum (SDC), and the consequences of allowing students to self-determine whether or not to attend class. The authors provided substantiating evidence that attending class enhances student performance. Accordingly, the school has elected to make student attendance a requirement. This is a welcome contribution to the field and will help the faculty at other medical schools provide evidence-based support when requiring student attendance. It will be interesting to follow how this change affects performance and the students’ reaction to the requirement.

·

Basic reporting

Backgroud:
Although authors describe about Early Clinical Experience (ECE), large group sessions and others about Shared Discovery Curriculum (SDC) in the text fairly in detail, it will be lot easier for readers to comprehend by showing the curriculum map.

Experimental design

Methods:
Description at lines 63-65 about the Progress Clinical Skills Examination (PCSE) needs more detail. Although they cite URL of the article by Gold et al., readers would like to know at least the number of the questions in the post encounter components.

Validity of the findings

Discussion:
 Please discuss about the conflicting results of the relationship between attendance and grades in the field of medical education in the previous researches.
 Authors address the limitations only about that this is not the experimental study. The data is the only one year is also the limitation of this study, although authors will not be able to accumulate the data because medical school’s decision to start requiring students attend the large group sessions.

Conclusions:
I believe that the conclusion to suggest causal relationship between large group sessions and students’ mastery of clinical applications of basic science knowledge is inappropriate.

Additional comments

The authors are to be congratulated on this piece of excellent work seeking the evidence for the newly introduced curriculum. This paper would be of interest for many medical educators. The methods are appropriate for this type of study. The paper is also clearly written and presented. However there are several minor points to be corrected.

---

## Round 0.2 · accepted · Accept

Hi David,

I am pleased to tell you that your manuscript has been accepted for publication in PeerJ.

Best regards,

Yoshi Marunaka

Reviewer 1 ·

Basic reporting

No Comment

Experimental design

No Comment

Validity of the findings

No Comment

Additional comments

Thank you for the revisions to the manuscript. I am pleased you found the comments useful.

·

Basic reporting

'no comment'

Experimental design

'no comment'

Validity of the findings

'no comment'

Additional comments

Authors’ revisions are well accepted. This paper would be of interest for the readers in the field of health professional education.